# Topologically frustrated dynamics of crowded charged macromolecules in charged hydrogels

Di Jia[1] & Murugappan Muthukumar[1]

Movement of charged macromolecules in crowded aqueous environments is a ubiquitous phenomenon vital to the various living processes and formulations of materials for health care. While study of diffusion of tracer amounts of probe macromolecules trapped inside concentrated solutions, gels, or random media has led to an enhanced understanding of this complex process, the collective dynamics of charged macromolecules embedded inside congested charge-bearing matrices still remains to be fully explored. Here we report a frustrated dynamics of DNA and synthetic polyelectrolytes inside a charged host hydrogel where the guest molecules do not diffuse. Instead, they exhibit a family of relaxation processes arising from a combination of conformational entropy and local chain dynamics, which are frustrated by the confinement from the gel. We also have developed a model explaining this new universality class of non-diffusive topologically frustrated dynamics of charged macromolecules.

---

[1] Department of Polymer Science and Engineering, University of Massachusetts, Amherst, MA 01003, USA. Correspondence and requests for materials should be addressed to M.M. (email: muthu@polysci.umass.edu)

The movement of charged macromolecules dispersed in a gel matrix is ubiquitous in crowded biological constructs throughout the human body and in applications such as drug delivery, therapeutic implants, and macromolecular separations[1–29]. It is well known that the solute macromolecules undergo diffusion and their measured diffusion coefficient is dictated by either their molecular weight or their physical size, depending on the nature of the physical constraints presented by the matrix[2–4]. While we observe diffusion at lower concentrations and molecular weights of the solute macromolecules, we find that their dynamics become frustrated and localized at higher concentrations and molecular weights when their radius of gyration $R_g$ is much greater than the mesh size $\xi$ of the matrix. As the concentration and molecular weight of the charged guest macromolecule are increased, there is a transition from the diffusional behavior to the non-diffusive localized dynamics. In the non-diffusive state occurring when the gel meshes are much smaller than the solute macromolecule, the conformations of the solute are entropically trapped preventing diffusion but experiencing dynamical fluctuations. Such a dynamical state of polyelectrolytes, theorized to be arising from entropic fluctuations of chain conformations inside the meshes of the hydrogel, exhibits a collective dynamics composed of a family of relaxation times. Although our observed law looks similar to the dynamical behavior observed in glassy systems[30–34], the present mechanism of entropic fluctuations under athermal conditions is distinctly different from molecular arrest created by lowering the temperature. The origin of this dynamical state lies in the topological chain connectivity of the macromolecules and hence this state is called topologically frustrated localized state and its dynamics as the topologically frustrated dynamics.

Traditionally, the essentials of diffusion of probe macromolecules in gels or other confining random media have been addressed by mainly three different mechanisms[2–11] (Fig. 1a–c). The first mechanism is the Ogston model[5], where the probe molecule, modeled as a rigid particle (Fig. 1a), collides against the matrix and diffuses slower than in the bulk, depending on the ratio of the particle size to the average mesh size of the confining medium. The second mechanism is the well known reptation model[6–9], where the macromolecule under strong entanglement constraints undergoes diffusion by an one-dimensional random walk of the chain contour (Fig. 1b). The third mechanism is based on entropic barriers created by the matrix with heterogeneous mesh size distribution for the dynamics of the polymer[10–12]. In this model portrayed in Fig. 1c, $R_g$ of the chain is comparable to the mesh size $\xi$ so that the chain diffusion occurs by negotiating essentially a single barrier. All of the above three mechanisms have been supported by experiments[13–29] in the order of weak, strong, and intermediate confinement conditions of the background matrix for the probe diffusion.

When a solute macromolecule undergoes diffusion in solutions or in a gel where the above three models are applicable, the time dependence of the averaged monomer density correlation function $<\rho_{\mathbf{q}}(t)\rho_{-\mathbf{q}}(0)>$ obeys the diffusion law of exponential decay[35,36],

$$<\rho_{\mathbf{q}}(t)\rho_{-\mathbf{q}}(0)> \sim e^{-\Gamma t}, \Gamma \equiv Dq^2, \qquad (1)$$

where $D$ is the diffusion coefficient, $q$ is the scattering wave vector ($q = (4\pi n/\lambda)\sin(\theta/2)$, with the scattering angle $\theta$, wave length of the incident monochromatic light $\lambda$, and refractive index of the medium $n$), and $t$ is the correlation time. The dynamical mode of density fluctuations is called diffusive if the two signatures of Eq. (1), namely, the exponential decay of the correlation function and the quadratic dependence of the decay rate $\Gamma$ on the scattering wave vector $q$ are satisfied. If these two signatures are not observed then the relaxation mode is non-diffusive. After demonstrating that Eq. (1) is valid, the diffusion coefficient is extracted and its dependence on the various characteristics of the host-guest systems is then established.

We report a non-diffusive dynamical behavior of charged macromolecules embedded in a charged hydrogel, displaying topologically frustrated localization driven by conformational

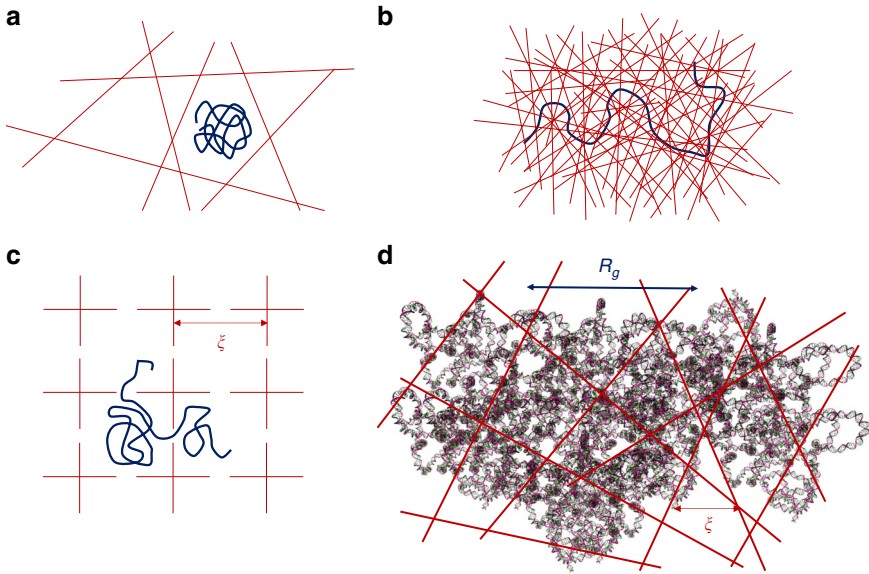

**Fig. 1** Three familiar models of macromolecular transport under confinement, and the topologically frustrated dynamical state. **a** Ogston's model, **b** Reptation, and **c** Entropic Barrier. **d** Cartoon of topological frustration when large guest macromolecules are crowded in a gel matrix. In **c**, where $R_g$ of the guest molecule is comparable to the mesh size $\xi$ of the gel, the molecule diffuses by negotiating essentially single entropic barriers. In **d**, $R_g$ is much bigger than the mesh size $\xi$ which itself is sufficiently big to hold large numbers of fluctuating monomers. Each macromolecule occupies tens of meshes with a family of local relaxation rates, and redistribution of portions of the molecule involves multiple entropic barriers which are strongly correlated. All of these barriers must be simultaneously breached by the guest molecule for it to move a distance comparable to its size. The required enormous time for this correlated process results in the non-diffusive topologically frustrated dynamics

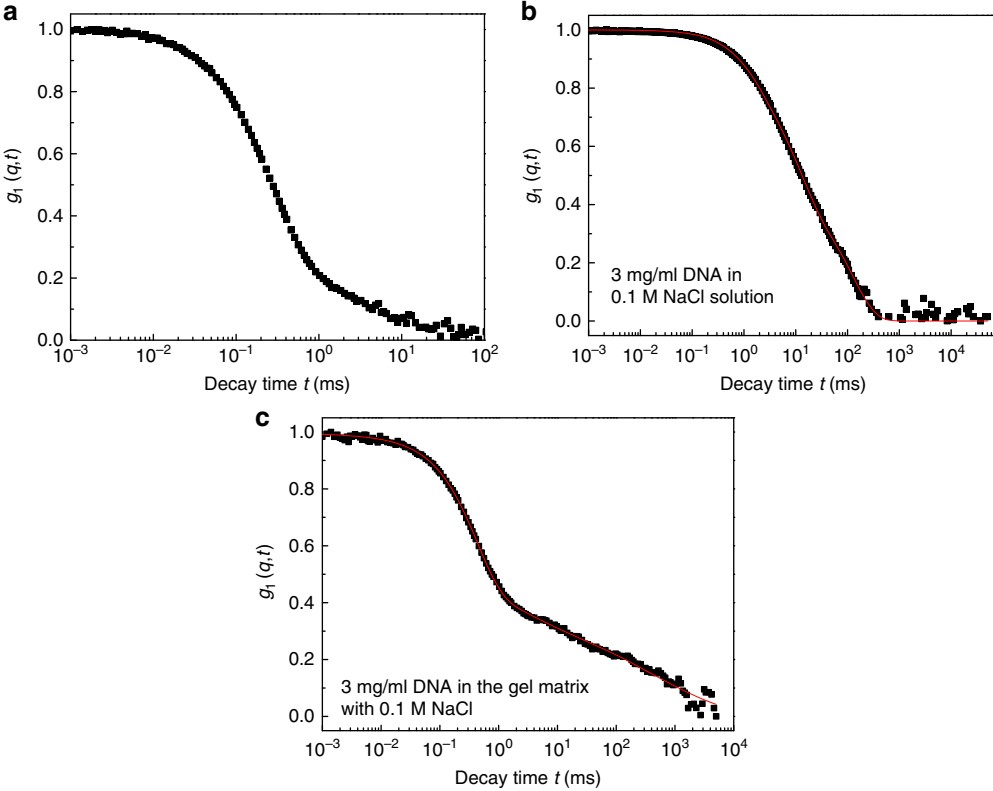

**Fig. 2** Typical normalized field-correlation function $g_1(q, t)$ at scattering angle 30°. **a** The host polyelectrolyte gel, **b** 3 mg/mL guest DNA molecules in solution and **c** 3 mg/mL guest DNA embedded inside the host hydrogel. Salt concentration for all samples is 0.1 M NaCl. The red lines are the best fits in **b**, **c**

entropy of the macromolecules. Our finding is based on dynamic light scattering (DLS) of deoxyribonucleic acid (DNA) and a synthetic polyelectrolyte (sodium polystyrene sulfonate, NaPSS) embedded in a well-controlled charged hydrogel. We do not observe Eq. (1) for higher concentrations and molecular weights of the guest polyelectrolytes residing inside a hydrogel. Instead, we report the following law,

$$<\rho_{\mathbf{q}}(t)\rho_{-\mathbf{q}}(0)> \sim e^{-(\Gamma t)^{1/3}}, \tag{2}$$

where $\Gamma$ is the decay rate. When the molecular weight and concentration of the guest molecules are low enough such that their spatial dimensions are comparable or smaller than the mesh size of the host gel, the dynamics naturally transforms into Eq. (1). In an effort to rationalize the observed effect, we present a mechanism in Fig.1d, along with a theory. When the size of the macromolecule is much larger than the mesh size and when the mesh size is large enough to hold substantial number of polymer segments ($R_g \gg \xi \gg \ell$, $R_g$ is the radius of gyration, $\xi$ is the mesh size, and $\ell$ is the monomer size), the chain is partitioned simultaneously into numerous entropic traps with a family of entropic barriers separating the traps. Inside each trap, the chain segments undergo their local dynamics. Due to a distribution in the mesh size, there are fluctuations in the number of segments partitioned into the meshes resulting in a family of relaxation times. As sketched in Fig. 1d, transfer of segments from one mesh into a new empty mesh cannot be an isolated process but it involves cooperative motion of all segments partitioned into many meshes separated by their numerous entropic barriers which are strongly correlated. This scenario is in contrast with the simple entropic barrier model (Fig. 1c), where essentially single barriers control chain diffusion. Under the condition of $R_g \gg \xi \gg \ell$ (Fig. 1d), all entropic barriers, which are strongly correlated over all occupying

meshes, must simultaneously be breached in order for the chain to move a distance comparable to its size. The time required for the center of mass of the chain to diffuse under such circumstances is impractically long due to the cooperativity among the entropic barriers. We call this entropically arrested situation as the topologically frustrated dynamical state. For this scenario of chain dynamics, we have derived a theory based on a superposition of varying confinement free energy associated with entropic fluctuations inside meshes and the underlying chain dynamics inside the meshes. In addition to explaining the experimental observations of Eq. (2), this theory opens a new universality class of topologically frustrated polymer dynamics.

## Results

**Representative correlation functions of DNA**. The emergence of the frustrated dynamics of charged macromolecules under confinement from a hydrogel is illustrated in Fig. 2. The electric field correlation function $g_1(q, t)$ from DLS, which is directly proportional to the monomer density correlation function $<\rho_{\mathbf{q}}(t)\rho_{-\mathbf{q}}(0)>$, is plotted against the delay time $t$ for (a) the host polyelectrolyte gel, (b) the guest DNA molecules in solution, and (c) the guest DNA embedded inside the host hydrogel. The host gel is the same in Fig. 2a, c. Even a visual inspection of the field correlation functions reveals that the dynamics of DNA crowded in the gel matrix (Fig. 2c) is qualitatively different from that in solution alone (Fig. 2b) and that the gel alone has its own characteristic dynamics. The above features for DNA are also observed for sodium poly(styrene sulfonate) as the guest charged macromolecule, as described below, demonstrating the generality of the observed phenomenon.

Analysis of such data as in Fig. 2 shows the following features: (a) The host gel has its intrinsic dynamics of elasticity with its

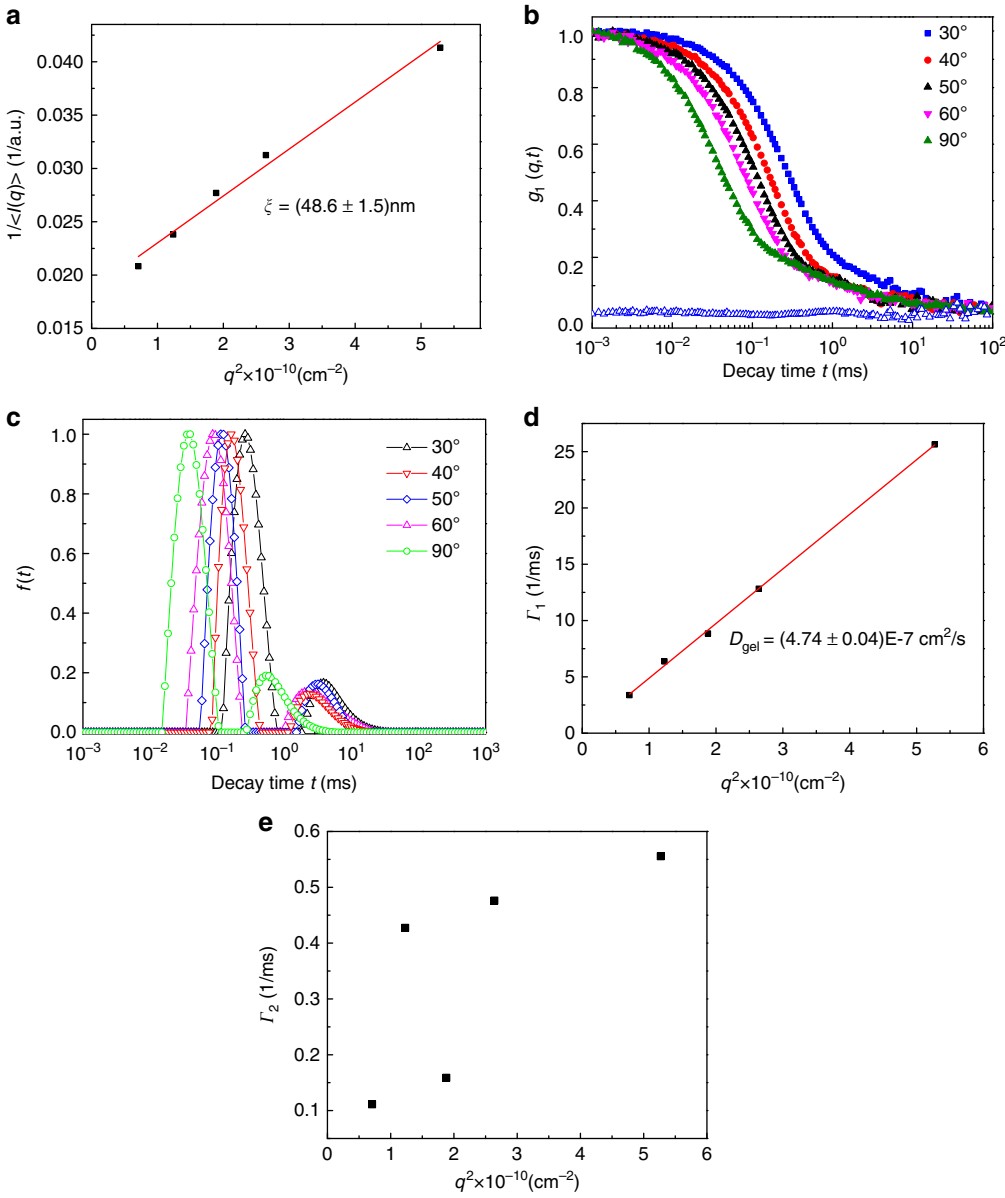

**Fig. 3** Characterization of the gel matrix. **a** The Ornstein-Zernike plot of the inverse average scattering intensity $1/I(q)$ measured by static light scattering of the host gel versus $q^2$, yielding the mesh size $\xi = 48.6 \pm 1.5$ nm. $I(q)$ is averaged over 3 independent measurements. **b** Normalized field-correlation function $g_1(q, t)$ at different scattering angles for the gel matrix with 0.1 M NaCl measured by dynamic light scattering. The blue open triangles are the residuals between the fitting curve and the original data of $g_1(q, t)$ at the scattering angle of 30° as an example. **c** Corresponding relaxation time distribution function obtained from CONTIN fit at different scattering angles for the gel matrix. **d** $q^2$ dependence of the relaxation rate $\Gamma_1$ of the first mode for the gel matrix. **e** Relaxation rate of the second mode $\Gamma_2$ as a function of $q^2$ for the gel matrix

elastic moduli measured by $g_1(q, t)$. (b) The guest polyelectrolyte molecules undergo diffusion in their solutions. (c) Under confinement inside the host gel, there is a coupling between the dynamics of gel elasticity and guest chains. The elastic moduli of the gel remain qualitatively the same as the pure gel, with only modest modifications made by the guest molecules, while the dynamics of the latter are significantly modified. The guest polyelectrolyte chains undergo a dynamical transition from diffusion into a topologically frustrated localization dynamics depending on the severity of the confinement. In the regime of diffusion, the monomer-monomer density correlation function (proportional to $g_1(q, t)$) is an exponential decay ($\sim e^{-\Gamma t}$, Eq. (1)), whereas in the topologically frustrated state it is a stretched exponential ($e^{-(\Gamma t)^{1/3}}$, Eq. (2)). The stretched exponential

behavior, which is totally unrelated to any thermal glassy-like behavior, arises from entropic fluctuations of the polyelectrolyte chains entrained by the meshes of the host hydrogel.

**Gel characterization and elastodynamics**. The host gel is poly (acrylamide-co-sodium acrylate) gel with 10% charge density and 0.2% crosslink density equilibrated in aqueous 0.1 M NaCl solution. The crosslink density was designed to be low in order to elicit the dynamics of very large charged guest macromolecules (with molar masses higher than $10^5$ Da, in comparison with our previous studies[28,37]).

Static light scattering: The mesh size $\xi$ of the equilibrated gel is obtained from the static light scattering intensity $I(q)$ at the

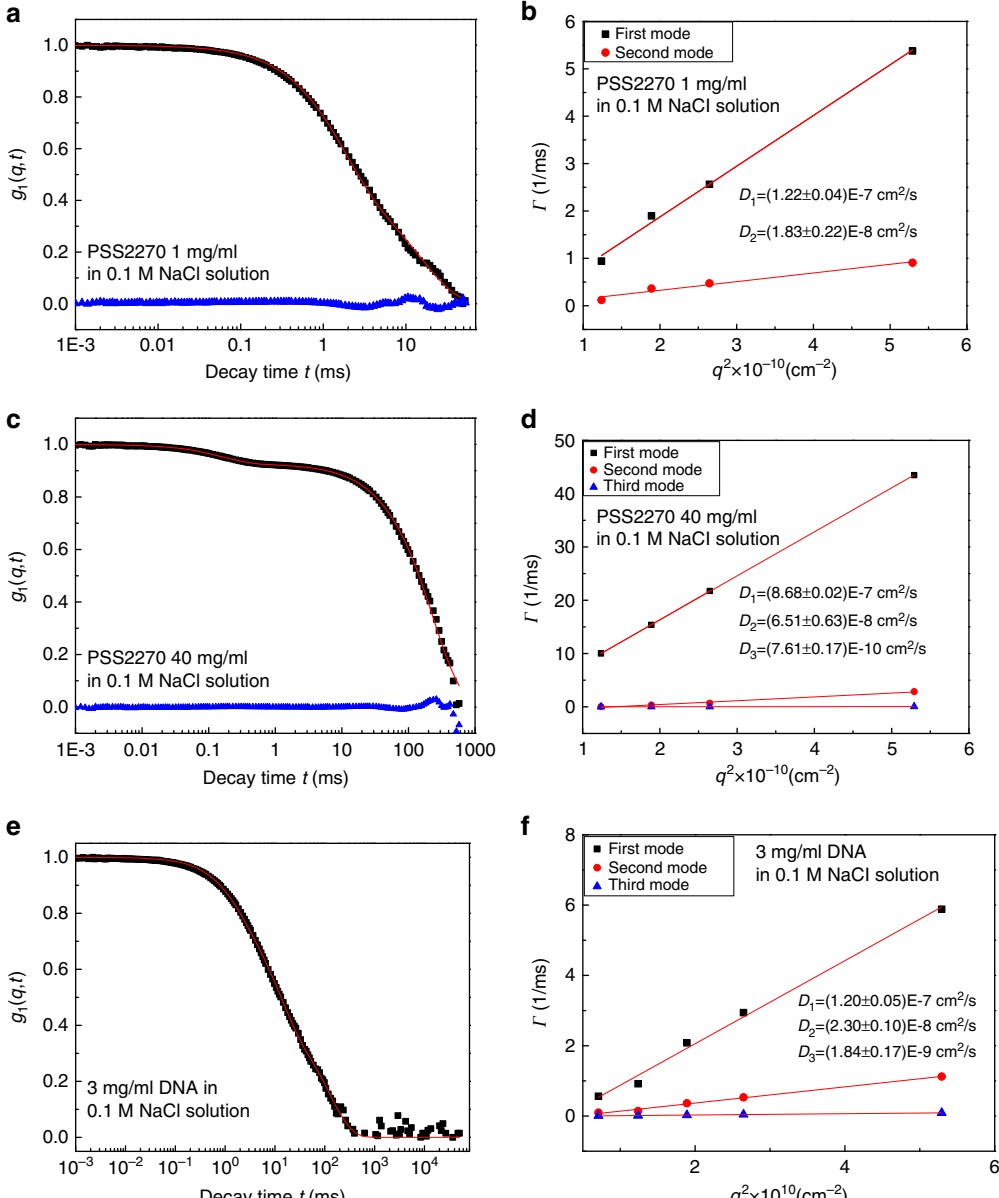

**Fig. 4** Dynamics of polyelectrolytes in solutions. **a**, **c**, **e** Normalized field-correlation function $g_1(q, t)$ at scattering angle 30° for (**a**) PSS2270 solution at $C = 1$ mg/mL, (**c**) PSS2270 solution at $C = 40$ mg/mL, (**e**) DNA solution at $C = 3$ mg/mL. The blue triangles are the residuals between the fitting curve and the original data. **b**, **d**, **f** Corresponding $q^2$ dependence of the relaxation rates $\Gamma$ for (**b**) PSS2270 solution at $C = 1$ mg/mL, (**d**) PSS2270 solution at $C = 40$ mg/mL, (**f**) DNA solution at $C = 3$ mg/mL, respectively. All samples are with 0.1 M NaCl

scattering wave vector $q$, and using the Ornstein-Zernike equation[36],

$$I(q) = \frac{I(q \to 0)}{(1 + q^2\xi^2)}. \qquad (3)$$

As shown in Fig. 3a, a plot of $1/I(q)$ versus $q^2$ is linear and the mesh size of the host gel is $\xi = 48.6 \pm 1.5$ nm.

Dynamic light scattering: The electric field correlation function $g_1(q, t)$ is proportional to the correlation function of the longitudinal component of the displacement $u_\ell(q, t)$ of gel strands, resolved into the $q$-th Fourier mode[38–44],

$$g_1(q, t) \sim \langle u_\ell(q, 0) u_\ell(q, t) \rangle = \langle u_\ell^2(q, 0) \rangle e^{-\Gamma_{gel} t}, \Gamma_{gel} \equiv D_{gel} q^2. \qquad (4)$$

where the gel diffusion coefficient $D_{gel} = (K + 4\mu/3)/\zeta$ ($K$ is bulk modulus, $\mu$ is shear modulus, and $\zeta$ is friction coefficient).

$g_1(q, t)$ at different scattering angles are given in Fig. 3b. These correlation functions are analyzed with several alternate fitting procedures such as CONTIN and multiple exponential fits[36,45]. As one typical example, the quality of the fit is illustrated in terms of the residuals for $\theta = 30°$ in Fig. 3b. An equivalent representation of data in Fig. 3b is provided in Fig. 3c as the distribution $f(t)$ of decay times $t$ for various scattering angles. The DLS of gels in general has a rich history exhibiting several modes of relaxation[41,46,47]. For our gel of low crosslink density, we find two modes, which are unrelated to the 'fast' and 'slow' modes described in the literature[48,49]. The most dominant mode contributes up to about 84% of the correlation function and its decay rate is proportional to $q^2$ as shown in Fig. 3d. Therefore this mode is diffusion-like and the corresponding elastic diffusion

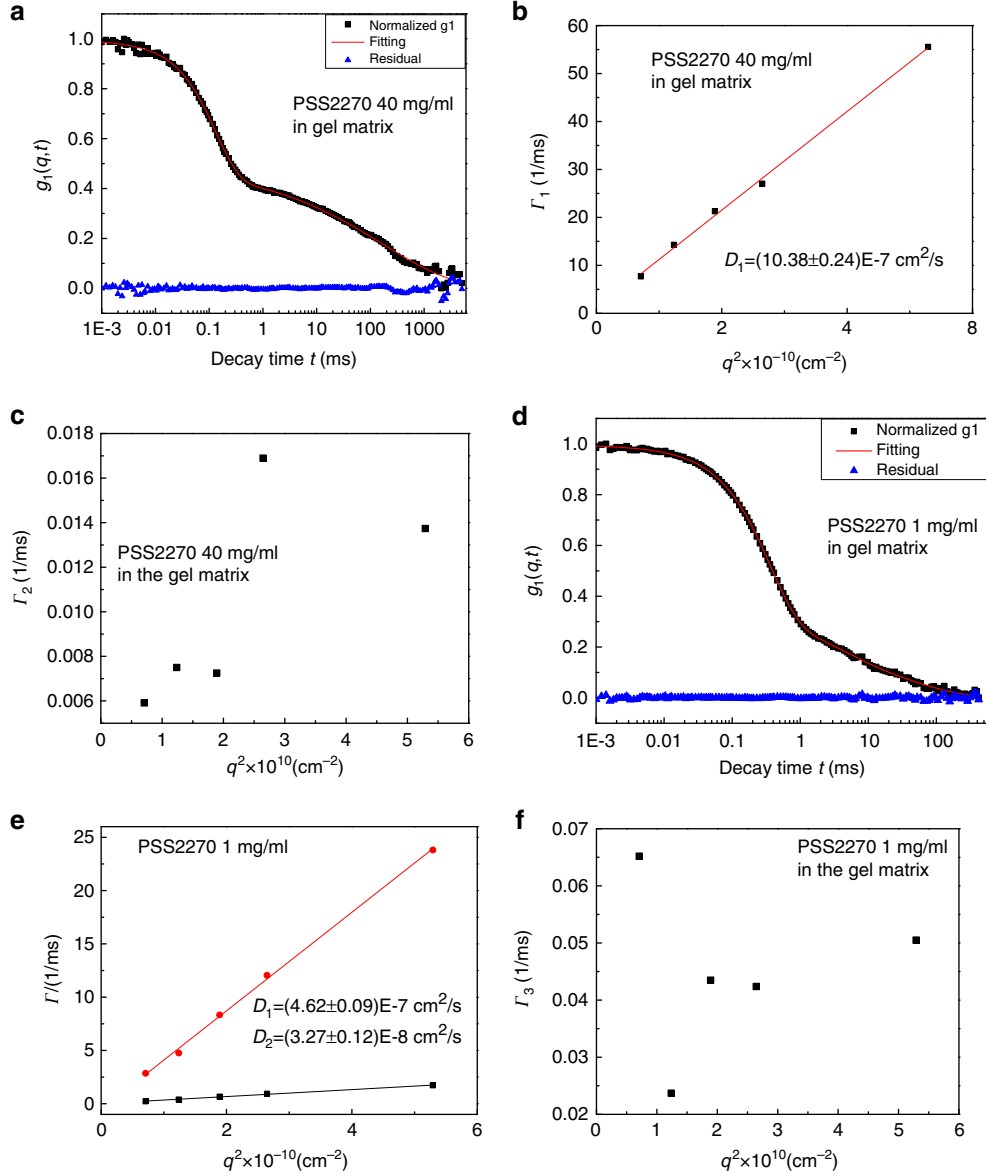

**Fig. 5** Dynamics of polyelectrolytes in gel matrix. **a**, **d** Normalized field-correlation function $g_1(q, t)$ of PSS2270 at 40 and 1 mg/mL, respectively, in gel matrix at scattering angle 30°. The blue triangles are the residuals between the original data (black) and the fitting curves (red). **b**, **c** Fitting results of $\Gamma$ vs. $q^2$ of different modes at all angles for PSS2270 at 40 mg/mL. **e**, **f** Fitting results of $\Gamma$ vs. $q^2$ of different modes at all angles for PSS2270 at 1 mg/mL. All samples are with 0.1 M NaCl

coefficient of the gel is

$$D_{gel} = (4.74 \pm 0.04) \times 10^{-7} \, \text{cm}^2/\text{s}. \qquad (5)$$

The other minor mode is non-diffusive, since the decay rates $\Gamma_2$ at all scattering angles do not have a linear relationship with $q^2$, as shown in Fig. 3e. We have analyzed all DLS data as homodyne since the heterodyne contribution[50–53] arising from inherent chemical inhomogeneity of our hydrogel with a very low crosslink density is weak (see Supplementary Fig. 1 and Supplementary Note 1) and it does not alter the conceptual conclusions of the report.

**Dynamics of guest polyelectrolytes in solutions**. The radius of gyration $R_g$ of polystyrene sulfonate (PSS) in dilute solutions at 0.1 M NaCl and room temperature was measured using static light scattering (Supplementary Fig. 2 and Supplementary Note 2). We have investigated five molar masses, $M_w = 126, 234,$

587, 1188, and 2270 kDa, and these samples are named as PSS126, PSS234, PSS587, PSS1188, and PSS2270, respectively. The values of $R_g$ are 16.4, 24.1, 58, 78.3, and 96.4 nm for PSS126, PSS234, PSS587, PSS1188, and PSS2270, respectively. The corresponding values of the overlap concentration $C^*$ ($= 3M_w/4\pi\mathcal{N}R_g^3$, with $\mathcal{N}$ = Avogadro number) of PSS are 11.32, 6.63, 1.2, 1.0, and 0.98 mg/mL, respectively. The Guinier plots used in obtaining $R_g$ were constructed at PSS concentrations lower than $C^*$. The representative field-correlation functions for NaPSS and DNA in 0.1 M NaCl solutions are given in Fig. 4, along with their corresponding decay rates.

The normalized $g_1(q, t)$ of PSS2270 at the guest polymer concentration $C = 1$ mg/mL is given in Fig. 4a. The best fit (red curve, Fig. 4a) for PSS2270 at $C = 1$ mg/mL is a double exponential, and the two rates (first and second) obey $\Gamma = Dq^2$ (Fig. 4b). The emergence of two diffusive modes and their values of the diffusion coefficient, $D_1 = (1.22 \pm 0.04) \times 10^{-7}$ cm²/s and $D_2 = (1.83 \pm 0.22) \times 10^{-8}$ cm²/s, are consistent with the previous

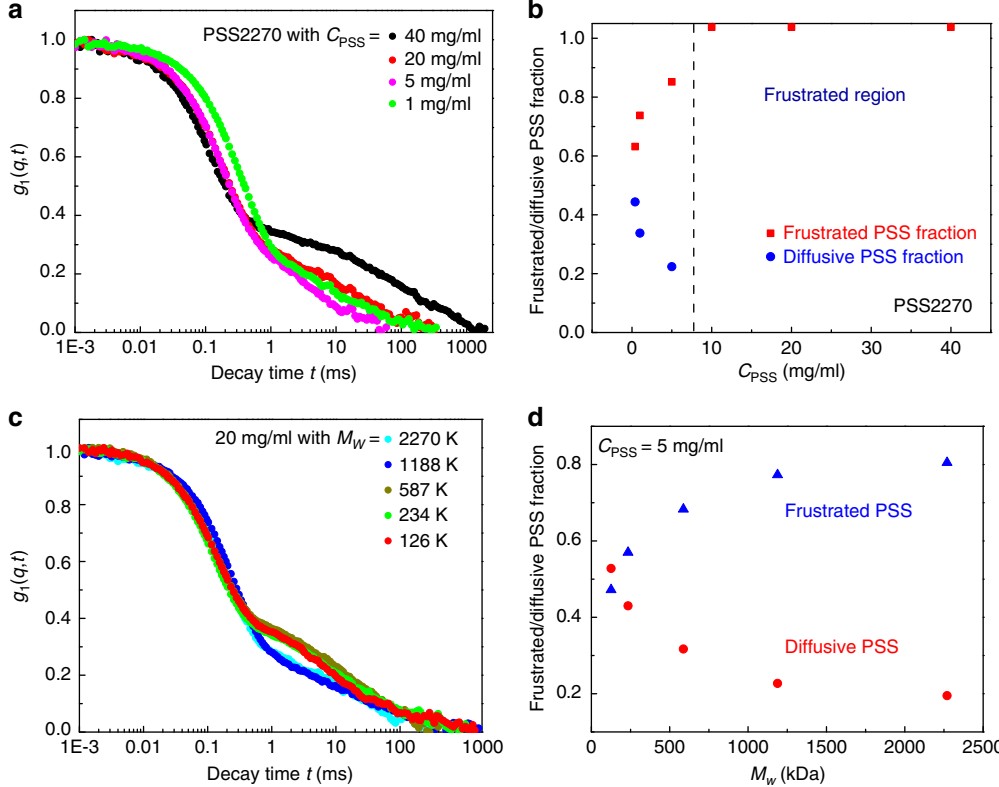

**Fig. 6** Dynamical transition of PSS in gel matrix with different $M_w$ and $C$. **a**, **c** Normalized field-correlation function $g_1(q, t)$ at scattering angle 30° of PSS in the gel matrix with 0.1 M NaCl for (**a**) PSS2270 with different PSS concentrations; (**c**) PSS with different $M_w$ at the same PSS concentration $C = 20$ mg/mL. **b** Fractions of frustrated and diffusive PSS chains for PSS2270 at different PSS concentrations. The dashed line denotes the crossover from totally frustrated dynamics to part of PSS being diffusive. **d** Fractions of frustrated and diffusive PSS chains for PSS with different molar masses at the same PSS concentration 5 mg/mL. The fraction of frustrated and diffusive PSS comes from $(1 - a_1 - a_2)/(1 - a_1)$ and $a_2/(1 - a_2)$, respectively, in Eq. (8)

literature[48,49]. For $C = 40$ mg/mL, three diffusive modes emerge (Fig. 4c, d). All three modes are diffusive with the corresponding diffusion coefficients as $(D_1 = 8.68 \pm 0.02) \times 10^{-7}$ cm²/s, $(D_2 = 6.51 \pm 0.63) \times 10^{-8}$ cm²/s, and $D_3 = (7.61 \pm 0.17) \times 10^{-10}$ cm²/s. While $D_1$ and $D_2$ correspond to the familiar two modes known as fast and slow modes in polyelectrolyte solutions, the observed third mode is new, which appears at such high concentrations of NaPSS. We also see three diffusive modes for DNA solutions at high concentrations, consistent with previous observations in DNA solutions[54]. At 3 mg/mL of DNA, the normalized $g_1(q, t)$ is given in Fig. 4e along with a three exponential fit (red curve). The corresponding rates are plotted against $q^2$ in Fig. 4f, where the linear relationship of $\Gamma - q^2$ clearly demonstrates the diffusive nature of these modes. The respective diffusion coefficients are $D_1 = (1.20 \pm 0.05) \times 10^{-7}$ cm²/s, $D_2 = (2.30 \pm 0.10) \times 10^{-8}$ cm²/s, and $D_3 = (1.84 \pm 0.17) \times 10^{-9}$ cm²/s. The origin of the third mode for DNA solutions and NaPSS solutions still remain unexplained[54]. Nevertheless, it must be emphasized that all relaxation modes observed in NaPSS and DNA solutions are diffusive.

**Dynamics of guest polyelectrolytes in gel matrix.** In contrast to the diffusional behavior in solutions (Fig. 4a–d), a non-diffusional phenomenon emerges when NaPSS is embedded in the host gel matrix, depending on its molar mass ($M_w$) and concentration $C$. The field-correlation function $g_1(q, t)$ for PSS2270 at $C = 40$ mg/mL with 0.1 M NaCl in the gel matrix is given in Fig. 5a. The distinctly different shape of $g_1(q, t)$ compared to Fig. 4c (for PSS2270 in solution) and Fig. 2a (gel alone) is clearly evident. The best fit of $g_1(q, t)$ in Fig. 5a can be accomplished by a sum of only

one exponential decay and a stretched exponential decay,

$$g_1(q, t) = a_1 e^{-\Gamma_1 t} + (1 - a_1) e^{-(\Gamma_2 t)^{\beta}}, \qquad (6)$$

where $\Gamma_1$ and $\Gamma_2$ are the decay rates for the two dynamical modes, $a_1$ ($=0.52$) is the weight of the first mode, and the value of the exponent $\beta$ ($=0.32$) is a measure of non-diffusive dynamical cooperativity of PSS chains inside the matrix. Analyzing $g_1(q, t)$ data at multiple scattering angles reveals that the first mode (exponential decay) is diffusional as defined in Eq. (1). This is demonstrated in Fig. 5b where $\Gamma_1 \sim q^2$ with the diffusion coefficient as $D_1 = (10.38 \pm 0.24) \times 10^{-7}$ cm²/s. Based on the data for the pure gel (Fig. 3d) and previous data on such gels, we can confidently assign the first mode as the elastic mode of the host gel ($D_1 = D_{gel}$), which is only modestly perturbed by the guest NaPSS molecules. Similar analysis shows that $\Gamma_2$ is not proportional to $q^2$ (Fig. 5c), indicating the non-diffusive nature of this stretched exponential mode and the value of $\beta$ is

$$\beta = 0.32 \pm 0.02. \qquad (7)$$

It is remarkable that the stretched exponential dynamics, which we ascribe to topological frustration, observed for the guest PSS2270 at 40 mg/mL under confinement by the host gel, is absent in solutions of PSS2270 at the same polymer concentration, and in fact it replaces all three diffusive modes seen in solutions (Fig. 4c, d). However, when $C$ is lowered to 1 mg/mL for PSS2270, an extra diffusive mode (in addition to the gel mode and

**Table 1 Number of diffusive modes in the system with different $M_w$ and $C$**

| $M_w$ (kDa) | 40 mg/mL | 20 mg/mL | 10 mg/mL | 5 mg/mL | 1 mg/mL | 0.4 mg/mL |
|---|---|---|---|---|---|---|
| 2270 | 1 | 1 | 1 | 2 | 2 | 2 |
| 1188 | 1 | 1 | 2 | 2 | 2 | 2 |
| 587 | 2 | 2 | 2 | 2 | 2 | 2 |
| 234 | 2 | 2 | 2 | 2 | 2 | 2 |
| 126 | 2 | 2 | 2 | 2 | 2 | 2 |

the stretched exponential mode) is seen. An example is shown in Fig. 5d, e, where the best fit is obtained by using

$$g_1(q,t) = a_1 e^{-\Gamma_1 t} + a_2 e^{-\Gamma_2 t} + (1 - a_1 - a_2)e^{-(\Gamma_3 t)^\beta}, \quad (8)$$

where $a_1$, $a_2$, and $a_3 = 1 - a_1 - a_2$ are the amplitudes of the three dynamical modes and $\Gamma_1$, $\Gamma_2$, and $\Gamma_3$ are the corresponding decay rates. The best fitted values are $a_1 = 0.67$, $a_2 = 0.1$, $a_3 = 0.23$, and $\beta = 0.39$. The analysis of $\Gamma_1$, $\Gamma_2$, and $\Gamma_3$ at multiple scattering angles shows that $\Gamma_1$ and $\Gamma_2$ are diffusive (Fig. 5e) and $\Gamma_3$ is non-diffusive since $\Gamma_3 - q^2$ plot is not linear (Fig. 5f) with the effective exponent $\beta = 0.39 \pm 0.03$ for the stretched exponential decay. The diffusion coefficient for the first mode is $D_1 = (4.62 \pm 0.09) \times 10^{-7}$ cm$^2$/s which is consistent with that for pure gel ($D_{gel} = (4.74 \pm 0.04) \times 10^{-7}$ cm$^2$/s), representing the elastodynamics of the host gel. So the first mode in Eq. (8) is the gel mode, while the second and third modes represent PSS dynamics: the second diffusive mode indicates that a part of the PSS is diffusive when PSS concentration is lowered to 1 mg/mL, and the third non-diffusive stretched exponential mode indicates that the rest of the PSS chains are still frustrated. The diffusion coefficient of the second mode is $D_2 = (3.27 \pm 0.12) \times 10^{-8}$ cm$^2$/s, which is an order of magnitude smaller than even the fastest diffusion coefficient ($(1.22 \pm 0.04) \times 10^{-7}$ cm$^2$/s) observed in solutions at the same $C = 1$ mg/mL. This reduction in the diffusion coefficient due to the presence of the host matrix is both expected and consistent with previous reports. Thus we see two coexisting modes (one diffusive and one non-diffusive frustrated dynamics) for the guest molecules at lower NaPSS concentrations. As described above, for $C = 40$ mg/mL, the diffusive mode of the guest chains is absent and the guest chain is entirely frustrated due to confinement by the host gel. Therefore, a dynamical crossover for NaPSS in the gel matrix from diffusion at lower $C$ and $M_w$ values to the topologically frustrated dynamics at higher $C$ and $M_w$ values is expected, as addressed below.

**Dynamical transition of guest polyelectrolytes inside the host gel.** The dynamical transition for PSS2270 is shown in Fig. 6a as a function of $C$. For $C = 20$ mg/mL and 40 mg/mL, the best fit for $g_1(q, t)$ is according to Eq. (6), with one diffusive gel mode and one non-diffusive stretched exponential mode corresponding to the frustrated localized dynamics. However, for $C$ below 5 mg/mL, an additional diffusive mode appears indicating the coexistence of some guest chains undergoing diffusion and the rest being localized by the confining host matrix. Thus there is a transition from fully frustrated dynamics at higher polyelectrolyte concentrations to a combination of frustrated dynamics and diffusion at lower values of $C$. The fraction of the frustrated dynamics, defined as $(1 - a_1 - a_2)/(1 - a_1)$, and the complementary fraction of diffusive dynamics of the guest chains are obtained from Eq. (8) and are portrayed in Fig. 6b for PSS2270 (the dynamical transition boundary indicated by the dashed line). Also, the dynamical transition occurs when the molar mass of the guest chain is varied from 2270 to 126 kDa at the fixed value of $C = 20$ mg/mL (Fig. 6c). For $M_w = 2270$ and 1188 kDa, there are only one diffusive gel mode and one non-diffusive mode of

frustrated dynamics of PSS (fit by Eq. (6)). On the other hand, when $M_w$ is 587 kDa or below, a second diffusive mode representing diffusion of the guest PSS inside the gel is observed (fit by Eq. (8)). Moreover, when $C$ is low, there are always diffusive PSS molecules. For example, for $C = 5$ mg/mL, the molar mass dependence of the fractions of frustrated and diffusive guest molecules as analyzed with Eq. (8) is given in Fig. 6d. Analogous to Fig. 6b, the mode of frustrated dynamics dominates at higher $M_w$ and the diffusive mode of the guest PSS becomes progressively significant as $M_w$ is decreased. The relative populations of these two modes depend on the combination of $C$ and $M_w$.

A summary of the number of diffusive modes in the whole system with different $M_w$ and $C$ is given in Table 1. The parameter space where only one diffusive mode corresponding to the gel dynamics is labeled as 1. Here the dynamics of the guest chains follows only the non-diffusive stretched exponential mode. Outside these 1 entries, "2" indicates that there are two diffusive modes, including one gel mode and one diffusive PSS mode in the system. The rest of PSS chains still obey the frustrated non-diffusive dynamics (Fig. 6b, d). The diffusion coefficient of the guest PSS (labeled as $D_2$) inside the matrix is given in Fig. 7a, with its corresponding numerical values in Fig. 7b. It is evident from Fig. 7a that the diffusive mode of PSS occurs only for lower values of $M_w$ and $C$. At higher values of $M_w$ and $C$, the guest chains follow overwhelmingly the frustrated dynamics. The transition between the two dynamical processes occurs at a lower $C$ if $M_w$ is higher. This is shown as the orange dashed curve in Fig. 7a. Since the inverse relation between the concentration and molar mass for the dynamical transition to occur is qualitatively similar to the behavior of the overlap concentration $C^*$ in solution, $C^*$ is also included as the dotted blue curve in Fig. 7a. As seen from Fig. 7a, the observed threshold $C - M_w$ curve for the onset of the dynamical transition inside the gel is significantly higher than the overlap concentration criterion in solutions. It should be noted in Fig. 7b that for $M_w = 126$, 234, and 587 kDa, $D_2$ decreases with $C$ whereas for $M_w = 1188$ and 2270 kDa, $D_2$ increases with $C$. These different trends in terms of $M_w$ values are attributed to the ratio of $R_g/\xi$. For lower molar masses, the chain dimension is comparable or smaller than $\xi$ and the diffusion is slowed down by the collisions of the guest chains against the host matrix, as reported earlier. On the other hand, at higher values of $R_g/\xi$, the cooperativity between the gel matrix and the guest chains results in a new dynamical behavior where $D_2$ increases with $C$, eventually leading to the topologically frustrated dynamical state. For the latter state occurring at higher $C$ and $M_w$ values, the dynamical exponent $\beta \simeq 0.32$. As shown in Fig. 7c for PSS2270 in the gel matrix, the effective value of $\beta$ increases slightly as $C$ is reduced, while the fraction of this stretched exponential mode decreases (Fig. 6b), as a representation of the crossover behavior between the topologically frustrated dynamics and diffusion. The topologically frustrated state is extremely long-lived during the experimentally relevant time scales and eventual diffusion is expected to arise for unrealistically long time scales as in other strongly frustrated thermal systems. In contrast to the two kinds of trends in Fig. 7b for the guest chains, the gel responds monotonically to increases in $C$ at different $M_w$ of the guest

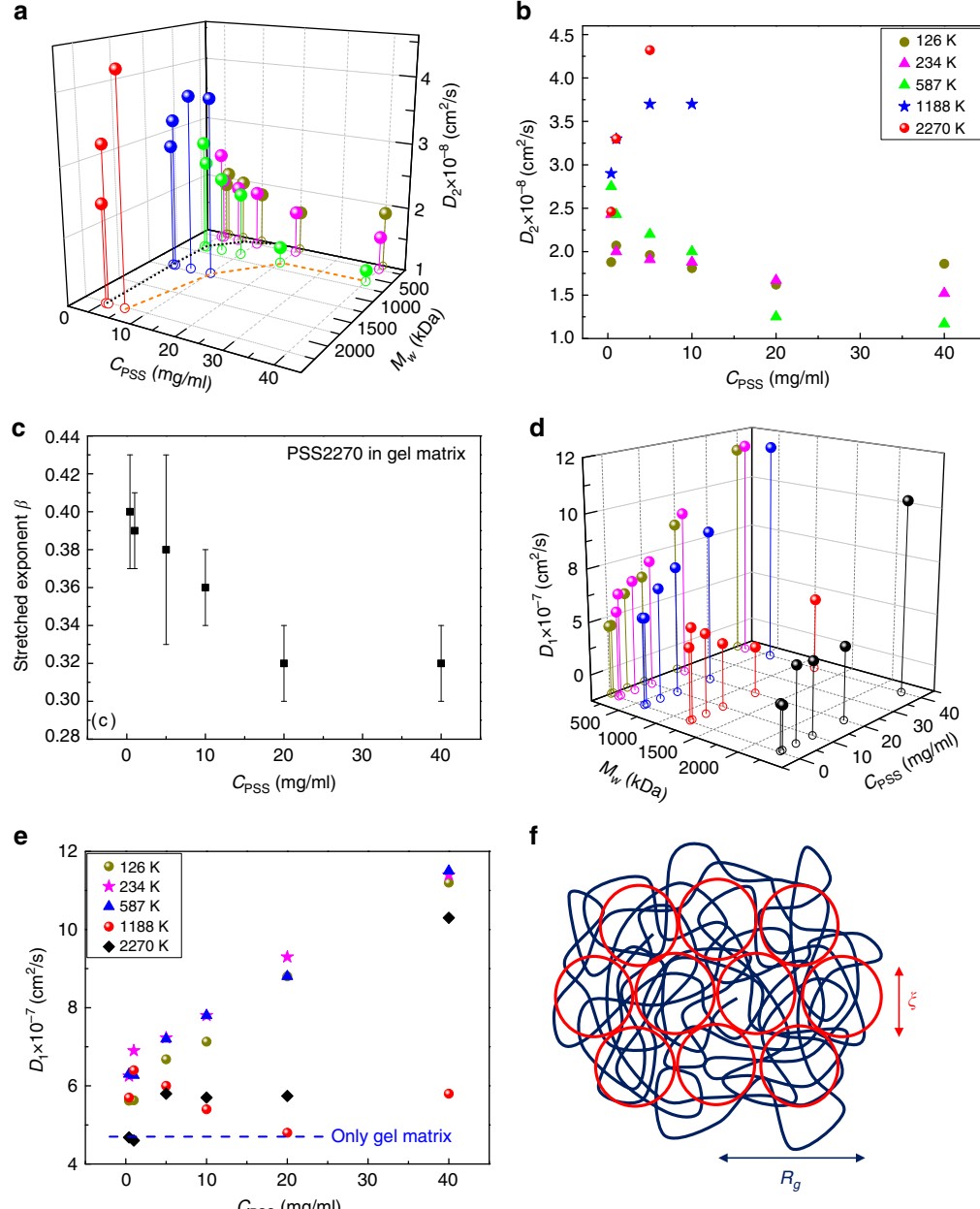

**Fig. 7** Diffusion coefficients and stretched exponential exponent. **a** Diffusion coefficient of the guest PSS ($D_2$) inside the gel matrix corresponding to the whole parameter space of Table 1. The unfilled circles are the projections of the values of $D_2$ denoting the corresponding coordinate values of $C_{PSS}$ and $M_w$. The orange dashed line is the boundary of dynamical transition for PSS in the gel matrix and the blue dotted line is the overlap concentration $C^*$ as a function of $M_w$ for PSS in solutions. **b** The corresponding numerical values of **a**. Same colors indicate the same $M_w$ in **a**, **b**. **c** Average value of $\beta$ at different PSS concentrations for PSS2270. Each of the data points denotes the average of three individual replicates, and error bars are ± sd. **d** Elastic diffusion coefficient of the gel ($D_1$) corresponding to the whole parameter space of Table 1. The unfilled circles are the projections of the values of $D_1$ denoting the corresponding coordinate values of $C_{PSS}$ and $M_w$. **e** The corresponding numerical values of **d**. The dashed blue line indicates the elastic diffusion coefficient ($D_1$) of the gel matrix without PSS. Same colors indicate the same $M_w$ in **d**, **e**. **f** Cartoon of partitioning of monomers of a single confined chain into a large number of meshes (in three dimensions)

molecules as shown in Fig. 7d, e, where the elastic diffusion coefficient of the gel ($\equiv D_1$) is provided in terms of $C$ and $M_w$. $D_1$ is higher than its value for the pure gel (dashed blue line in Fig. 7e) and it increases with $C$, consistent with earlier results.

Augmenting the above observation on the synthetic polyelectrolyte (NaPSS) as the guest molecule, the phenomenon of topologically frustrated dynamics occurs also for biological polyelectrolytes such as DNA as introduced in Fig. 2c. The best fit of $g_1(q, t)$ in Fig. 2c for DNA inside the gel is with Eq. (8),

where

$$g_1(q, t) = 0.53e^{-\Gamma_1 t} + 0.06e^{-\Gamma_2 t} + 0.41e^{-(\Gamma_3 t)^{0.32}}, \qquad (9)$$

where $\Gamma_1 = 2.3 \times 10^3 \, \text{s}^{-1}$, $\Gamma_2 = 2.2 \times 10^2 \, \text{s}^{-1}$, and $\Gamma_3 = 2.59 \, \text{s}^{-1}$. The analysis of DLS data on the decay rates at different scattering angles shows that $\Gamma_1$ and $\Gamma_2$ are diffusive with the corresponding diffusion coefficients $D_1 = 4.4 \times 10^{-7} \, \text{cm}^2/\text{s}$ and $D_2 = 5.2 \times 10^{-8} \, \text{cm}^2/\text{s}$ (Supplementary Fig. 3a). $D_1$ is the elastic

diffusion coefficient of the gel, and $D_2$ is the diffusion coefficient for the diffusive part of DNA inside the matrix. This value of $D_2$ is an order of magnitude smaller than the fastest diffusion coefficient ($1.27 \times 10^{-7}$ cm$^2$/s) observed in solutions at the same DNA concentration. On the other hand, the third mode is a stretched exponential with its rate $\Gamma_3$ being non-diffusive as seen in the nonlinear relationship of $\Gamma_3$ versus $q^2$ (see Supplementary Fig. 3b). In fact, for DNA concentration of 3 mg/mL, this mode representing the frustrated localization dynamics contributes 87% ($=0.41 \times 100/(0.41 + 0.06)$), with the rest being the diffusional mode. The best fit at different scattering angles gives the averaged $\beta$ as

$$\beta = 0.31 \pm 0.03, \tag{10}$$

which is equivalent to the value observed for the PSS system, demonstrating the ubiquity of the topologically frustrated dynamics of polyelectrolytes in a confining hydrogel.

**Theory**. As documented above, the frustrated dynamics of the guest chains emerges as the dominant process for very large values of $C$ and $M_w$. Under these congested polyelectrolyte concentrations, each chain pervades many meshes (Fig. 1d). $R_g$ of the guest molecule with high $M_w$ can be much larger than the mesh size $\xi$ ($=48.6$ nm) so that each chain occupies many meshes. For example, when $R_g/\xi$ is about 2, one chain of PSS2270 occupies about 34 meshes ($\sim (4/3)\pi(R_g/\xi)^3$ in three dimensions) even when the embedded chains are in isolation (Fig. 7f). At higher polymer concentrations, due to chain interpenetration, the cooperativity of segments belonging to a single chain, but partitioned among tens of meshes, is even more pronounced. However, since the gel mesh is large enough ($\sim 50$ nm) to contain considerable number of polymer segments, and the host hydrogel has $\sim 96\%$ water (by weight fraction), the segments inside each mesh can undergo considerable conformational fluctuations. As the mesh size is much larger than the segment length, the polyelectrolyte chains are not confined in a tube-like environment invoked in the context of the reptation dynamics (Fig. 1b), and hence entanglement effects may be ignored. The various meshes function as entropic traps and the collective migration of chains can occur only if all intervening entropic barriers between the traps (meshes) are simultaneously negotiated. Every time a group of segments, which are in one of the meshes on the surface of the region inside the gel that is occupied by a chain, tries to move outward into a new mesh, the rest of the partitions of the chain exerts an entropic pulling force resisting the move. As a result, the time required for all partitions to move a distance comparable to the chain size is enormous relative to experimental time scales and therefore the center of mass of the chain is essentially localized. However, due to finite temperature and since the hydrogel contains enormous amount of water, the chain segments inside the meshes are not dynamically frozen. They undergo their local Rouse-Zimm dynamics[4], with the additional effect from fluctuations in the number of segments inside the various meshes. When the molar mass is lower than a certain threshold value, $R_g/\xi$ becomes smaller and the usual diffusional dynamics is released. Consequently, a transition from the topologically frustrated dynamics to diffusion occurs as $M_w$ is decreased. As the experimental observations show, a threshold concentration of the guest polyelectrolyte (inversely related to $M_w$) is also a crucial control parameter in promoting the formation of the topologically frustrated state. At higher values of $C$, the probability of finding an empty mesh for filling it by the segments in an adjacent mesh is lower and hence the topologically frustrated state is extremely long-lived. As $C$ and $M_w$ are reduced, the entropic barrier

landscape becomes smoother allowing simultaneous presence of both dynamical processes.

To understand the relation between the reduction in conformational entropy associated with the segments confined in meshes and the observed stretched exponential dynamics, let $m$ segments be in one mesh. The confinement free energy[3] $F_{\text{con } f}$ of these segments is proportional to $m$,

$$\frac{F_{\text{conf}}}{k_B T} = A_\xi m, \tag{11}$$

where $A_\xi$ depends on $\xi$ and the chain statistics inside the mesh ($k_B T$ is the Boltzmann constant times the absolute temperature). Therefore, the probability of having $m$ segments inside a mesh of size $\xi$ follows from the Gibbs distribution, $P(m) \sim \exp(-F_{\text{con}}/k_B T)$, as[3]

$$P(m) \sim e^{-A_\xi m}. \tag{12}$$

The dynamics of the chain segments inside the mesh is expected to obey the Rouse dynamics[4] due to screening of hydrodynamics, intra-chain excluded volume and electrostatic interactions. Since the longest relaxation time $\tau_R$ for the Rouse dynamics is proportional to the square of the contour length of the strand, the characteristic relaxation rate $\Gamma_m$ for the strand with $m$ segments is[4]

$$\Gamma_m = \frac{A_R}{m^2}, \tag{13}$$

where $A_R$ is a prefactor from the lowest Rouse mode for polymer chains. If there are no fluctuations in the number of segments inside a mesh such that $m$ is a fixed number for each mesh, the field-correlation function is given by

$$g_1(t) = e^{-\Gamma_m t}. \tag{14}$$

In reality, the number of segments inside a mesh is a fluctuating quantity and its probability is given by Eq. (12). Therefore, the experimentally observed $g_1(t)$ is a superposition of all realizations of $m$ given by Eq. (12) with Eq. (13) being the decay rate for each value of $m$ so that we get

$$g_1(t) = \int dm P(m) e^{-\Gamma_m t}. \tag{15}$$

Substituting Eqs. (12) and (13) in Eq. (15), and performing the integral with the saddle point approximation[55] (Supplementary Note 3), we obtain

$$g_1(t) = e^{-(\Gamma t)^{1/3}}, \tag{16}$$

where the rate $\Gamma = (27/4)A_\xi^2 A_R$. Thus, the combination of the Rouse dynamics and free energy penalty associated with entropic fluctuations inside meshes leads to the observed experimental result of $\beta \simeq 1/3$. In general, for a chain with an effective size exponent $v$, where $v$ is the size exponent ($R_g \sim M_w^v$), $\beta = 1/(2v + 2)$ in the Rouse regime (see Supplementary Note 3), which is applicable to even semiflexible chains. The value of the exponent in Eq. (16) depends on whether the Rouse dynamics[4] or the Zimm dynamics[4] is used to describe the strand dynamics inside the meshes. If the Zimm dynamics is used, the theoretical value of the exponent $\beta$ is $1/(3v + 1)$ (Supplementary Note 3). For good solvents $\beta = 5/14 \simeq 0.357$, and for Gaussian chains $\beta = 0.4$, instead of 1/3 based on the Rouse dynamics. As the concentration of the guest chains is reduced, the hydrodynamic screening is less and the strand dynamics would move from the Rouse behavior toward the Zimm behavior. As a consequence of this crossover

behavior, the value of the exponent $\beta$ is expected to slightly increase from the asymptotic value of 1/3, as seen in Fig. 7c.

## Discussion

In summary, when charged macromolecules are confined inside a similarly charged host gel matrix, a non-diffusive topologically frustrated dynamics emerges for higher concentrations and molecular weights. The origin of this phenomenon lies in the topological correlations of the guest chain resulting from conformational fluctuations inside the meshes of the gel under the conditions of $R_g \gg \xi \gg \ell$. As $C$ and $M_w$ are lowered, the frustrated guest molecule progressively transforms into diffusion. Experimental observations of DLS in both DNA and NaPSS embedded in the gel reveal that the monomer density correlation function for the frustrated state obeys the stretched exponential law with the exponent $\beta \simeq 0.32$. These results are interpreted using a theory based on fluctuations in the number of segments occupied inside the meshes and the Rouse dynamics in each mesh. The dynamics is driven entirely by conformational entropy and is unrelated to any temperature dependent effects. Inclusion of partial screening of hydrodynamics and segmental interactions increases the value of $\beta$, but still keeping within a narrow range, $2/5 > \beta > 1/3$, instead of the full range $(0 < \beta < 1)$ seen in temperature-dependent glassy and aging systems. The range of $\beta$ for the topologically frustrated dynamics is also outside the value expected in incoherent scattering studies[36]. We should note that even for situations where the solute molecule is permanently but sparsely anchored to the gel matrix[56–58], the topologically frustrated dynamics is expected to be valid. The present discovery opens a new universality class of polyelectrolyte dynamics. Further issues such as the estimation of lifetime of the long-lived topologically frustrated state, role of electrostatics, dynamics of labeled guest chains, and whether or not the same phenomenon is present even in uncharged polymer systems under the condition $R_g \gg \xi \gg \ell$, naturally emerge for future investigations.

## Methods

**Materials**. Acrylamide (40% w/v) and Bis(acrylamide) (2% w/v) solutions were bought from Amresco and used without further purification. Tetramethylethylenediamine (TEMED), ammonium persulfate (APS), and sodium acrylate were purchased from Sigma-Aldrich and used as received. Sodium polystyrenesulfonate (NaPSS) of five molecular weights (126, 234, 587, 1188, 2270 kDa) was purchased from ScientiÞc Polymers (catalog numbers 624, 625, 626, 628, and 923). The solutions were prepared using deionized water and sodium chloride. The Hydrophilic Polyvinylidene Fluoride (PVDF) filters with two pore sizes 220 and 450 nm were purchased from Millex Company.

**Gel synthesis and sample preparation**. Poly(acrylamide-co-sodium acrylate) (PAM-PAA) gel was synthesized using free radical polymerization[37]. 0.163 g acrylamide, 0.022 g sodium acrylate and 0.78 mg Bis(acrylamide) were mixed with certain amount of concentrated PSS solutions, then 0.1 M NaCl solution was added until the total volume is 5 mL. The sample was bubbled with nitrogen gas for 15 min to remove any dissolved oxygen which can inhibit the reaction. After that, 7.5 μL TEMED and 2.5 mg APS were added to initiate the reaction. In order to do light scattering measurement, the pregel solution was quickly filtered (200 or 450 nm PVDF filter) into a light scattering tube to remove the dust and then was continued to be bubbled with nitrogen. The reaction was continued for 2 h at room temperature. The crosslinking density of the gel is 0.2% (molar ratio of the crosslinker to the total moles of the monomers and crosslinker) and the charge density is 10% (molar ratio of the charged monomer to the total monomers). After the reaction, the filtered solution of NaPSS (same concentration as inside the gel) was added to the tube in excess amount on top of the gel to maintain equilibrium between the gel and the solution. The samples were kept at room temperature for 3 weeks to reach equilibrium before measurement.

**DNA sample preparation**. High molecular weight calf thymus DNA (Sigma D1501) was dissolved in 10 mM Tris-Cl and 1 mM EDTA buffer solution with 0.1 M NaCl (pH 7.8 at 20 °C). Then the samples were kept at 4 °C for 3 days to fully dissolve before use. The approximate average length of DNA was determined by agarose gel electrophoresis to be 15 kbp. Sample preparation of DNA for DLS measurement is the same as NaPSS. We have confirmed that DNA is not

chemically cross-linked to the gel matrix (see Supplementary Fig. 4 and Supplementary Note 4).

**Dynamic light scattering (DLS)**. DLS measurement was performed on a commercial spectrometer equipped with a multi-τ digital time correlator (ALV-5000/E) using a wavelength of 514.5 nm laser light source. DLS measures the intensity-intensity time correlation function $g_2(q, t)$ by means of a multi-channel digital correlator and related to the normalized electric field correlation function $g_1(q, t)$ through the Siegert relation[36]. For each sample, the intensity at each of the scattering angles 30°, 40°, 50°, 60°, and 90° was correlated, and the relaxation time averaged for three different spatial locations within the samples. CONTIN method[45] and multiple exponential fitting method[43,44] were used to analyze the characteristic relaxation rate $\Gamma_i$ at each angle. Based on $D_i = \Gamma_i/q^2$, diffusion coefficient was calculated.

**Data fitting of stretched exponential functions**. For the correlation function of the gel and PSS mixture system, multiple modes were first confirmed by CONTIN method. In the presence of multiple modes and in order to facilitate comparison to a theory, the normalized $g_1(q, t)$ was fitted by a sum of one or two exponential decays and a stretched exponential function. $g_1(q, t)$ was fitted using ORIGIN software by minimization of error between the fitted prediction and the data. Iterations were performed until the best fitting curve was obtained within the tolerance limit. The residuals, which are obtained from the difference between the original data and the fitting curve, are randomly distributed about the mean of zero and do not have systematic fluctuations about their mean.

**Data availability**. The data that support the findings of this study are available from the corresponding author upon request.

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

## Acknowledgements

We thank the National Science Foundation (DMR-1504265), AFOSR Grant FA9550-17-1-0160, and the National Institutes of Health (Grant No. R01HG002776-11) for financial support.

## Author contributions

M.M. conceived the project. D.J. designed the experiments, synthesized the gel, and performed all experiments. M.M. developed the theory. Both authors discussed and contributed to the interpretation of the data. The whole project was supervised by M.M.

## Additional information

**Competing interests:** The authors declare no competing interests.

