## [Peer Review File · Nature Communications]

Reviewers' comments:

Reviewer #1 (Remarks to the Author):

This manuscript presents experimental evidence and theoretical modeling of a non-diffusive state observed with crowded charged molecules whose radius of gyration is greater than the mesh size of the charged hydrogel. When hydrogel meshes are much smaller than the charged molecule, the conformation is entropically trapped, preventing diffusion but experiencing dynamic fluctuations.

Methods, interpretations and conclusions appear to be solid. The concept is logical and appears to be novel, but the manuscript was difficult to follow for a non-expert in this field. This is of particular concern because this report of topologically frustrated dynamics could potentially be of significant interest to those in the fields of drug delivery, tissue engineering and protein/polymer separations.

Questions:

1. Having all the non-diffusive mode data in the supplementary information made it difficult to follow the main text. Further explanation would help clarify how the mode was determined as non-diffusive. For example, it is not clear on modes versus homodyne/heterodyne. Modes are fast or slow; fast relating to osmotic pressure, slow due to inhomogeneities in gel. This work presents a third mode due to topographically frustrated transport. Is each mode either homodyne/heterodyne?

2. Supplementary figures 1 and 4 show non-diffusive mode by comparing $\Gamma - q^2$ but Supplementary Figure 3 used $\ln\Gamma - \ln q$, why was this figure different?

3. Other papers have talked about multiple modes including diffusive modes and stretched exponential functions with gels. For example, the paper below models the system based on a reputation model not an entropy trap model. Is this relevant?

Zhang, E., et al. (2018). Viscoelastic behaviour and relaxation modes of one polyamic acid organogel studied by rheometers and dynamic light scattering. *Soft Matter*, 14(1), 73-82. doi:10.1039/c7sm02185b

Reviewer #2 (Remarks to the Author):

This paper describes a rigorous polymer physical analysis of guest DNA and synthetic polyelectrolyte (polystyrene sulfonate, PSS) molecules embedded in a host charged hydrogel system of poly(acrylamide-co-sodium acrylate). The movement of charged guest molecules within charged gel systems is of course an important biological problem. This is a system studied by the authors previously (references 28 & 37) but here the focus is on dynamic processes of high molecular weight guest molecules (>105 Da). To accommodate these large guest molecules the hydrogels in this study were prepared at much lower crosslink density (0.2%) than the author's previous studies (~2-4%).

My main concern is with the interpretation of the non-diffusive mode in their system as being topologically frustrated. Prior studies have shown that encapsulation of proteins or DNA with in-situ forming hydrogels often leads to irreversible interactions and chemical crosslinks between the polymerizing polymer chains and the encapsulated protein or DNA. A few examples include doi.org/10.1023/B:PHAM.0000003368.66471.6a, DOI: 10.1016/j.jconrel.2004.03.001 and doi.org/10.1007/s11095-005-9395-x. Incomplete DNA or protein release from these in-situ formed hydrogels is often observed. Prior studies of free radically polymerized hydrogels suggest this incomplete release is due to radical crosslinking of the DNA or proteins to the hydrogel network.

The novel nondiffusive mode described in these experiments arise at very large molecular weights or high polymer concentrations. Both conditions, at the same weight ratios, would lead to higher monomer concentrations thus increasing the probability of the guest PSS or DNA to be chemically crosslinked to the in situ formed hydrogel system formed. Based on these prior hydrogel release studies, I would suggest the system under study in this manuscript it likely not topologically frustrated but rather immobilized through radical crosslinking the polymer network. Naïvely I would imagine that the scaling theory outlined by Jia and Muthukumar might still be valid with the ends of the DNA or PSS chains crosslinked to the hydrogel being able to undergo conformational fluctuations within the gel meshes following Rouse-Zimm dynamics as assumed for a topologically frustrated system.

Minor comments

- (1) there were a few minor errors related to the figures. For instance, bottom of page 11 they state the shape of normalized field-correlation function of 5a compared to 4b – but they mean 4c.
- (2) I found several of the figures difficult to read. Text and data points in many figures could easily be made larger in Figures 2-6. I found the 3-D plots in Figure 7 difficult to see.
- (3) In figure 6c, the 5 PSS chains seem to show two types of behavior (basically the lowest 3 MWs look identical in the normalized field-correlation functions and highest 2 MWs look identical), yet the fractions of frustrated/diffusive systematically change with molecular weight. Why?
- (4) Why does the persistence length of the chains not play a role in this non-diffusive dynamic? If PSS or DNA are fluctuating inside the meshes, I would have anticipated a slower Rouse dynamic for the stiff DNA chains compared to the flexible PSS chains within the meshes.

Reviewer #3 (Remarks to the Author):

This manuscript provides strong experimental and theoretical evidence for the existence of a new regime of restricted diffusion of macromolecules trapped within cross-linked gels. I have two major questions, which may result from my own limitations in understanding of this specific field. The first arises at the bottom of page 3. I cannot personally vouch for the statement: "In all of the previous studies of probe diffusion inside a gel, the time dependence of the averaged monomer density correlation function $\langle \rho_q(t)\rho_q(0) \rangle$ (measured by DLS) obeys the diffusion law of exponential decay^{35,36}, $\langle \rho_q(t)\rho_q(0) \rangle \sim e^{-\Gamma t}$ " That may be true, but it is a very broad statement that supports the high degree of novelty of this work "a new universality class of topologically frustrated polymer dynamics.". I simply do not know if that is true, but I would not bring it up if I did not have my doubts. Figure 1d is insufficiently clear to illustrate the new mechanism the authors are proposing so I do not really know if their proposed mechanism is different from Figure 1c, for example. My second general question stems from the following statement: "The general premise of our study is the movement of charged macromolecules dispersed in a gel matrix ..." I do not see what any of this has to do with charge. The authors use the term charge or charged twice in the title of this manuscript but I see nothing in Figure 1d, insofar as I can understand it, that has to do with charge ... I think this idea has to do with length scales. So, if the authors could clarify these two central ideas for me, I would be favorably inclined toward this manuscript.

Response to Referee #1

We thank the reviewer for the enthusiastic endorsement and raising an excellent point of improving the readability by even ‘non-experts’. We have addressed all points raised by the reviewer as detailed below. The comments of the reviewer are in black and our response is in blue.

This manuscript presents experimental evidence and theoretical modeling of a non-diffusive state observed with crowded charged molecules whose radius of gyration is greater than the mesh size of the charged hydrogel. When hydrogel meshes are much smaller than the charged molecule, the conformation is entropically trapped, preventing diffusion but experiencing dynamic fluctuations. Response: We thank the reviewer for this succinct summary and for appreciating the critical condition for the reported new state of non-diffusive dynamic fluctuation arising from correlated entropic trapping, namely the radius of gyration is greater than mesh size. We have emphasized this condition in the revised version (caption of Fig.1d). Also we have contrasted this dynamics with the uncorrelated entropic barrier model of Fig.1c.

Methods, interpretations and conclusions appear to be solid. The concept is logical and appears to be novel, but the manuscript was difficult to follow for a non-expert in this field. This is of particular concern because this report of topologically frustrated dynamics could potentially be of significant interest to those in the fields of drug delivery, tissue engineering and protein/polymer separations.

Response: We are grateful for the warm endorsement on novelty of our findings and for the nice suggestion. We agree that the paper must be reasonably easy to follow for even non-experts in this field, especially since our report of topologically frustrated dynamics is of significant importance in diverse fields of drug delivery , tissue engineering, and separation science. We have made a serious effort to make the paper readable by even ‘non-experts’. The changes are throughout the manuscript, particularly in the Introduction section.

Questions:

1. Having all the non-diffusive mode data in the supplementary information made it difficult to follow the main text. Further explanation would help clarify how the mode was determined as non-diffusive. For example, it is not clear on modes verses homodyne/heterodyne. Modes are fast or slow; fast relating to osmotic pressure, slow due to inhomogeneities in gel. This work presents a third mode due to topographically frustrated transport. Is each mode either homo-

dyne/heterodyne?

Response: We thank the reviewer for these suggestions to improve the readability of the paper. Specifically, our responses are as follows:

(a) We have moved all non-diffusive mode data from the previous supplementary information to the main text. The previous Supplementary figures, Figs. 1a, 3a, and 3b, are now Figs. 3e, 5c, and 5f, respectively.

(b) We define a mode to be diffusive if it satisfies two criteria: first, the field correlation function must decay exponentially with the correlation time, and second, the rate of decay must be quadratic in the scattering wave vector. If these two criteria are not met, then the mode is non-diffusive. These are clearly spelled out in the revision as on page 5.

(c) In our study, all modes are analyzed as homodyne. In our previous studies on gels of higher crosslink density, we had systematically investigated homodyne and heterodyne (originating from crosslink heterogeneity) analyses. In the present study, where the crosslink density is low, the heterodyne contribution is weak. We have illustrated this by showing heterodyne analysis only for the gel in the Supplementary Information. The difference between the homodyne and heterodyne analyses lies only in differences in the particular values of the diffusion coefficient, but not in whether the relaxation mode in our system is diffusive or non-diffusive. This is now clearly stated on pages 9-10.

(d) The rich literature on ‘fast’ and ‘slow’ modes of polyelectrolyte solutions at very low ionic strengths is not pertinent to the present system of a gel with embedded polyelectrolyte chains at relatively high ionic strength. Therefore we have now refrained from referring to our two diffusive modes as ‘fast’ and ‘slow’ to avoid any confusion; instead, we simply call these as ‘first’ and ‘second’ modes. The change is made throughout the paper.

2. Supplementary figures 1 and 4 show non-diffusive mode by comparing $\Gamma - q^2$ but Supplementary Figure 3 used $\ln\Gamma - \ln q$, why was this figure different?

Response: As recommended by the reviewer, we now present $\Gamma - q^2$ plots in Figs. 5c and 5f in the main text. In the original version we presented the data as a double logarithmic plot to emphasize that the diffusive law $\Gamma \sim q^2$ is not observed, that is, the slope of $\ln\Gamma$ versus $\ln q$ is not 2.

3. Other papers have talked about multiple modes including diffusive modes and stretched exponential functions with gels. For example, the paper below models the system based on a reputation model not an entropy trap model. Is this relevant?

Zhang, E., et al. (2018). Viscoelastic behaviour and relaxation modes of one polyamic acid organogel studied by rheometers and dynamic light scattering. *Soft Matter*, 14(1), 73-82. doi:10.1039/c7sm02185b

Response: DLS on the gel alone has been an active field for several decades, showing the appearance of multiple modes along with certain explanations specific to the particular gel. In our system, we observe one dominant diffusive mode and our current focus is on the effect of the gel in topologically frustrating a large macromolecule embedded inside it. Therefore, the reference suggested by the referee is not directly pertinent to the present report. Nevertheless, we have now given reference to Zhang et al (Reference 48), along with other key references (References 46 and 47) for DLS in gels alone.

Response to Referee #2

We thank the reviewer for endorsing the manuscript based on experimental rigor, novelty of the described phenomenon, and broad potential impact even in areas dealing with encapsulation of DNA and proteins. In addition, the reviewer raises whether there is immobilization of the guest DNA during the synthesis of the gel, while recognizing that even if this were to be true, the proposed new dynamics might still be valid. We have addressed all points raised by the reviewer as detailed below. The comments of the reviewer are in black and our response is in blue.

This paper describes a rigorous polymer physical analysis of guest DNA and synthetic polyelectrolyte (polystyrene sulfonate, PSS) molecules embedded in a host charged hydrogel system of poly(acrylamide-co-sodium acrylate). The movement of charged guest molecules within charged gel systems is of course an important biological problem. This is a system studied by the authors previously (references 28 & 37) but here the focus is on dynamic processes of high molecular weight guest molecules (>105 Da). To accommodate these large guest molecules the hydrogels in this study were prepared at much lower crosslink density (0.2%) than the authors previous studies ($\sim 2-4\%$).

My main concern is with the interpretation of the non-diffusive mode in their system as being topologically frustrated. Prior studies have shown that encapsulation of proteins or DNA with in-situ forming hydrogels often leads to irreversible interactions and chemical crosslinks between the polymerizing polymer chains and the encapsulated protein or DNA. A few examples include doi.org/10.1023/B:PHAM.0000003368.66471.6a, DOI: 10.1016/j.jconrel.2004.03.001 and doi.org/10.1007/s11095-005-9395-x. Incomplete DNA or protein release from these in-situ formed hydrogels is often observed. Prior studies of free radically polymerized hydrogels suggest this incomplete release is due to radical crosslinking of the DNA or proteins to the hydrogel network. The novel nondiffusive mode described in these experiments arise at very large molecular weights or high polymer concentrations. Both conditions, at the same weight ratios, would lead to higher monomer concentrations thus increasing the probability of the guest PSS or DNA to be chemically crosslinked to the in situ formed hydrogel system formed. Based on these prior hydrogel release studies, I would suggest the system under study in this manuscript it likely not topologically frustrated but rather immobilized through radical crosslinking the polymer network.

Response: Stimulated by the reviewer's comment on whether DNA is permanently cross-linked to the gel matrix, we have conducted two sets of new experiments. These experiments conclusively show that DNA is not cross-linked to the gel in our system, as described below.

Before we describe these new experiments, we would like to note that our observation of topologically frustrated state is universal as exhibited by both polystyrene sulfonate (PSS) and DNA. As PSS is known not to suffer from any acryl-based free radicals, it does not crosslink to the gel. Given that we observe the same phenomenon for both PSS and DNA, it is unlikely that DNA is crosslinked to our gel under our preparation conditions (also as demonstrated below). Furthermore, for DNA (as well as PSS), we observe simultaneous occurrence of both diffusive and stretched exponential behaviors under certain concentrations. It is unlikely that some DNA molecules escaped from cross linking while others participated exclusively in cross-linking. This fact also suggests that chemical bonding between DNA and our gel is unlikely. We would like to also note that we have used ammonium persulfate as the initiator under anaerobic conditions for synthesizing the gel in the presence of DNA, whereas the protocol used in the references mentioned by the reviewer is with a photo initiator and UV radiation. Nevertheless, we have taken the reviewer's question seriously and performed the following experiments.

(a) **Experiment 1:** We synthesized two samples of PAM-PAA hydrogels, one without DNA and the other with DNA, using the same experimental protocol described in the paper. Small pieces of these samples were then subjected to gel electrophoresis in agarose gel with voltage of 110 V for 1.5 hours. The gel electrophoresis data are shown in Fig.1 below (which is Supplementary Fig.4a), where the reference lane 1 is for the PAM-PAA gel without DNA, and lane 3 is for the PAM-PAA gel with 3 mg/mL DNA. Under the electric field, PAM-PAA gel piece is stuck inside the well and cannot move, but if the gel piece contained DNA then the DNA would be released if it is not chemically cross-linked to the gel. It is evident from the result of lane 3 that DNA has escaped from the PAM-PAA gel matrix in the presence of an electric field, demonstrating that DNA is not covalently bonded to the PAM-PAA gel matrix.

(b) **Experiment 2:** We synthesized two PAM-co-PAA solutions using the same protocol as in the paper, except now in the absence of the cross-linking agent. In the absence of the cross-linker, the formation of PAM-co-PAA linear chains is still a free radical polymerization. The first sample is PAM-co-PAA polymers mixed with DNA during synthesis. The second sample was obtained by first synthesizing PAM-co-PAA polymers without DNA and after the reaction ended, the resultant polymer solution was mixed with DNA. The DLS data on these two samples are shown in Figs. 2c and 2d (which are Supplementary Figs.4c and 4d), respectively, where the DNA concentration is 1mg/mL for both samples. If DNA can form chemical bonds with the polymer chains by free radicals, an extra ultra-slow mode is expected to show up in DLS, which we did not observe. The field correlation function g_1 for these two samples at the scattering angle of 30° is given in Fig.2b

FIG. 1: Supplementary Figure 4(a): Gel electrophoresis of pieces of PAM-co-PAA gels without DNA (lane 1) and with 3 mg/mL DNA (lane 3) using the standard agarose gel with 0.1 M NaCl at 110 V for 1.5 hours. DNA escapes from the PAM-co-PAA gel and moves in the agarose gel under the electric field.

(which is Supplementary Fig.4b), showing that these are essentially identical. Besides, as seen from Supplementary Figs.4c and 4d, there are three diffusive modes for both cases and the corresponding diffusion coefficients are quite similar, independent of whether DNA was present or not during the free radical polymerization of the monomers. Therefore, DNA is not chemically cross-linked to the polymer chains in our system.

Thus, based on the results from the above two sets of experiments, we can confidently conclude that there is no DNA immobilization due to cross linking between DNA and the gel matrix in the present system and that the topologically frustrated state arises from uncrosslinked DNA and PSS confined inside the matrix. Now we have added the two sets of new experiments in the Supplementary Fig.4 and Supplementary Note 4.

Naïvely I would imagine that the scaling theory outlined by Jia and Muthukumar might still be valid with the ends of the DNA or PSS chains crosslinked to the hydrogel being able to undergo conformational fluctuations within the gel meshes following Rouse-Zimm dynamics as assumed for a topologically frustrated system.

FIG. 2: Supplementary Figure 4: (b) Field correlation function g_1 at scattering angle of 30° for 1 mg/mL DNA mixed with the PAM-co-PAA solutions before and after synthesis of the polymer without cross linker. (c) Plot of Γ versus q^2 for PAM-co-PAA polymer solution mixed with DNA before synthesis and (d) Plot of Γ versus q^2 for PAM-co-PAA polymer solution mixed with DNA after synthesis.

Response: This is a very insightful comment and we thank the reviewer for this excellent suggestion. In response, we have included the following sentence on page 22 along with the new references 57-59. “We should note that even for situations where the solute molecule is permanently but sparsely anchored to the gel matrix⁵⁷⁻⁵⁹, the topologically frustrated dynamics is expected to be valid.”

Minor comments

(1) there were a few minor errors related to the figures. For instance, bottom of page 11 they state the shape of normalized field-correlation function of 5a compared to 4b - but they mean 4c.

Response: Thanks. This typo is corrected.

(2) I found several of the figures difficult to read. Text and data points in many figures could easily be made larger in Figures 2-6. I found the 3-D plots in Figure 7 difficult to see.

Response: We have redrawn the figures and improved their resolution to enable easier read. Now the 3-D plots are easier to see and the figure captions are more detailed.

(3) In figure 6c, the 5 PSS chains seem to show two types of behavior (basically the lowest 3 MWs look identical in the normalized field-correlation functions and highest 2 MWs look identical), yet the fractions of frustrated/diffusive systematically change with molecular weight. Why?

Response: Please note that Fig.6c is for PSS concentration of 20 mg/mL, whereas it is 5 mg/mL in Fig.6d. No change is made in the revision.

(4) Why does the persistence length of the chains not play a role in this non-diffusive dynamic? Is PSS or DNA are fluctuating inside the meshes, I would have anticipated a slower Rouse dynamic for the stiff DNA chains compared to the flexible PSS chains within the meshes.

Response: As mentioned by the reviewer, chain stiffness (appearing as the effective size exponent ν) does indeed play a role in the non-diffusive dynamics. The value of the stretched exponential exponent β is $1/(2\nu + 2)$ in the Rouse regime and $\beta = 1/(3\nu + 1)$ in the Zimm regime. As already discussed in the paper, the value of β is in a narrow range around $1/3$ for both flexible and semiflexible chains ($3/5 < \nu < 1$) under both dilute and semidilute conditions. The general expression for β for the Zimm regime was already given in the earlier version and now the value of $\beta = 1/(2\nu + 2)$ for the Rouse regime is included in both the main text (on page 21) and Supplementary Note 3.

Response to Referee #3

We thank the reviewer for endorsing the manuscript and raising a few comments to improve the clarification of the presentation. We have addressed all points raised by the reviewer as detailed below. The comments of the reviewer are in black and our response is in blue.

This manuscript provides strong experimental and theoretical evidence for the existence of a new regime of restricted diffusion of macromolecules trapped within cross-linked gels. I have two major questions, which may result from my own limitations in understanding of this specific field. The first arises at the bottom of page 3. I cannot personally vouch for the statement: “In all of the previous studies of probe diffusion inside a gel, the time dependence of the averaged monomer density correlation function $\langle \rho q(t)\rho q(0) \rangle$ (measured by DLS) obeys the diffusion law of exponential decay^{35,36}, $\langle \rho q(t)\rho q(0) \rangle \sim e^{-\Gamma t}$ ” That may be true, but it is a very broad statement that supports the high degree of novelty of this work “a new universality class of topologically frustrated polymer dynamics.”. I simply do not know if that is true, but I would not bring it up if I did not have my doubts.

Response: In view of the reviewer’s remark, we have reworked this confusing sentence with additional explanation on pages 4-5. In our earlier version, the only purpose of this sentence was to convey that the signature of time evolution of the density correlation function is exactly exponential for any diffusion process including the diffusion of probe molecules inside a gel. We did not mean to state that diffusion of the probe was always observed. However, diffusion of probe molecules inside gels is generally observed including our own previous studies, but not in the present class of topologically frustrated dynamics.

Figure 1d is insufficiently clear to illustrate the new mechanism the authors are proposing so I do not really know if their proposed mechanism is different from Figure 1c, for example.

Response: We agree with the reviewer that the caption of Figs.1c and 1d is not sufficiently clear. We have now clearly explained the difference between the two mechanisms on page 5 and extensively in the caption of Fig.1. We have also added more description of Fig.1c on page 3. Briefly, in the entropic barrier mechanism represented in Fig.1c, where the chain size is comparable to the mesh size, chain diffusion is controlled by essentially isolated and uncorrelated entropic barriers. Hence the characteristic time for a chain to diffuse is determined by essentially a single entropic barrier. On the other hand, in the case of topologically frustrated dynamics corresponding to the condition where the polymer size is much larger than the mesh size, multiple correlated barriers must be simultaneously crossed (Fig.1d) for a chain to move a distance comparable to its size. This

time is so enormous that the chain is localized among many meshes allowing only local dynamical fluctuations, resulting in the new class of dynamics.

My second general question stems from the following statement: “The general premise of our study is the movement of charged macromolecules dispersed in a gel matrix ...” I do not see what any of this has to do with charge. The authors use the term charge or charged twice in the title of this manuscript but I see nothing in Figure 1d, insofar as I can understand it, that has to do with charge ... I think this idea has to do with length scales. So, if the authors could clarify these two central ideas for me, I would be favorably inclined toward this manuscript.

Response: We fully agree with the reviewer that it is indeed the length scales that leads to this phenomenon. We have now explicitly mentioned this throughout the paper, especially in the Introduction section. We have used the word ‘charged’ in the title to be specific about the present investigation where both the guest molecule and the host matrix are charged. Also, we are interested in polymer dynamics relevant to crowded biological systems where almost everything is charged. Furthermore, our previous theoretical work on uncharged polymers under confinement (J.Chem. Phys., 89, 2435, (1988); J. Chem. Phys., 90, 4594 (1989)) shows that uncharged chains can be induced to undergo self-attraction. This additional feature is expected to lead to other effects beyond the purely repulsive (athermal) case of the charged host-guest system studied here. In view of the ubiquity of charged systems in biological contexts and to accurately qualify the systems investigated here, we have kept ‘charged’ in the title. Nevertheless, we hope that the present work would stimulate further investigations on uncharged systems as well in exploring the general validity of the topologically frustrated localized dynamics. Now, we have also added “... and whether or not the same phenomenon is present even in uncharged polymer systems under the condition $R_g \gg \xi \gg \ell$, naturally emerge for future investigations” on page 22.

REVIEWERS' COMMENTS:

Reviewer #1 (Remarks to the Author):

The authors did a very nice job in addressing all concerns noted in the first review.

Reviewer #2 (Remarks to the Author):

The authors have addressed the majority of my concerns.

Reviewer #3 (Remarks to the Author):

The authors have made a very thorough, constructive, and specific response to all of the reviewers' comments. I am satisfied that the current version of the manuscript now merits publication.

Response to Referees

Referee #1:

The authors did a very nice job in addressing all concerns noted in the first review.

Referee #2:

The authors have addressed the majority of my concerns.

Referee #3:

The authors have made a very thorough, constructive, and specific response to all of the reviewers' comments. I am satisfied that the current version of the manuscript now merits publication.

We thank all reviewers for the endorsement.